# Molecular Cloning and Functional Analysis of *DXS* and *FPS* Genes from *Zanthoxylum bungeanum* Maxim

**DOI:** 10.3390/foods11121746

**Published:** 2022-06-14

**Authors:** Lu Tian, Jingwei Shi, Lin Yang, Anzhi Wei

**Affiliations:** 1College of Forestry, Northwest A&F University, Yangling, Xianyang 712100, China; t1anlu@nwafu.com.cn (L.T.); sjw@nwafu.com.cn (J.S.); yanglin123@nwafu.com.cn (L.Y.); 2Research Centre for Engineering and Technology of Zanthoxylum State Forestry Administration, Yangling, Xianyang 712100, China

**Keywords:** *Zanthoxylum bungeanum* Maxim., *ZbDXS*, *ZbFPS*, terpene, expresssion

## Abstract

*Zanthoxylum bungeanum* Maxim. (*Z. bungeanum*) has attracted attention for its rich aroma. The aroma of *Z**. bungeanum* is mainly volatile terpenes synthesized by plant terpene metabolic pathways. However, there is little information on *Z. bungeanum* terpene metabolic gene. In this study, the coding sequence of *1-deoxy-D-xylulose-5-phosphate synthase* (*DXS*) and *farnesyl pyrophosphate synthase* (*FPS*) were cloned from *Z. bungeanum* cv. ‘Fengxiandahongpao.’ *ZbDXS* and *ZbFPS* genes from *Z. bungeanum* with CDS lengths of 2172 bp and 1029 bp, respectively. The bioinformatics results showed that *Z. bungeanum* was closely related to citrus, and it was deduced that *ZbFPS* were hydrophilic proteins without the transmembrane domain. Subcellular localization prediction indicated that *ZbDXS* was most likely to be located in chloroplasts, and *ZbFPS* was most likely to be in mitochondria. Meanwhile, the 3D protein structure showed that *ZbDXS* and *ZbFPS* were mainly composed of α-helices, which were folded into a single domain. In vitro enzyme activity experiments showed that purified proteins *ZbDXS* and *ZbFPS* had the functions of *DXS* enzyme and *FPS* enzyme. Transient expression of *ZbDXS* and *ZbFPS* in tobacco significantly increased tobacco’s terpene content. Moreover, *ZbDXS* and *ZbFPS* were expressed in different tissues of *Z. bungeanum*, and the relative expression of the two genes was the highest in fruits. Therefore, this suggests that *ZbDXS* and *ZbFPS* are positively related to terpene synthesis. This study could provide reference genes for improving *Z. bungeanum* breeding as well as for the Rutaceae research.

## 1. Introduction

Secondary metabolites are essential organic matter during plant growth and development [1]. The secondary metabolic pathways of plants have self-protection mechanisms against various environmental threats such as environmental changes and biotic stress [2]. In particular, volatile organic compounds (VOCs) are one of the most widely studied substances, giving plants a variety of odors that play an essential role in plant growth and development [3]. VOCs are usually divided into terpenes, benzenoid aromatics, and fatty acids [4]. Terpenes account for the largest proportion of VOCs and are the most widely studied substances [5].

Terpenes are widely distributed in the plant field and are synthesized from isoprene (C_5_). According to the number of carbon atoms, they can be divided into monoterpenes (C_10_), sesquiterpenes (C_15_), and diterpenes (C_20_), triterpenes (C_30_), and polyterpenes. At present, the metabolic pathways of terpenes have been well interpreted. There are two metabolic pathways: the methylerythritol phosphate (MEP) pathway, which occurs in the plastid, and the mevalonate (MVA) pathway, which occurs in the cytosol [6]. The MEP pathway uses glyceraldehyde-3-phosphate and pyruvate as precursors to synthesize monoterpenes and diterpenoids by modifying various enzymes in the downstream pathway [7]. The MVA pathway uses acetyl-CoA as a precursor to synthesize sesquiterpenes [8].

The synthesis and accumulation of different types of terpene depend on the action of critical enzymes in the metabolic pathway. These enzymes are generally at the branching points of the synthetic pathway, catalyzing the formation of various precursors and intermediates. Therefore, the synthesis of aroma can be studied by cloning analysis and expression regulation of crucial enzyme genes [9]. Among them, 1-deoxy-D-xylulose-5-phosphate synthase (*DXS*) and farnesyl pyrophosphate synthase (*FPS*) are crucial enzymes in the upstream and downstream pathways of the pathway and play an essential role in terpene synthesis. *DXS* is the first crucial rate-limiting enzyme in the MEP pathway; It catalyzes pyruvate and glyceraldehyde 3-phosphate to 1-deoxyxylulose-5-phosphate [10]. The *DXS* family is a very tiny family currently known to have three subfamilies [11]. The *DXS1* family is mainly involved in the synthesis of isoprenoids required for primary metabolites, the *DXS2* family is thought to be involved in plant-specific isoprenoid synthesis, and the *DXS3* family is involved in hormone synthesis [12,13]. Related studies show that overexpression of *DXS* can affect plants. For instance, in the study of overexpression of the *DXS* gene in *Populus trichocarpa*, it was found that the *DXS* gene improves the tolerance of *Populus trichocarpa* to abiotic, biotic stress and is widely involved in plant growth and development, and physiological processes of stress [14]. In *geranium*, isolation and functional verification of its *DXS* gene (*GrDXS*), and overexpression of *GrDXS* in *geranium* and *Withania somnifera* by genetic transformation, showed that *GrDXS* overexpression results in increasing essential oils in *geranium* and withanolides in *W. somnifera* [15]. *FPS* is mainly based on GPP and IPP for catalytic synthesis of FPP, which is an essential rate-limiting step in terpene metabolism related to downstream terpenoid modification and plant stress resistance [16]. Silencing the FPS gene of Arabidopsis thaliana produces a greening phenotype associated with changes in chloroplast morphology and structure, and results in significant changes in cytoplasmic and cytoplasmic isoprene types. [17]. *DXS* and *FPS* are more widely studied in flowers and some model plants. However, Rutaceae plants with well-developed oil glands and unique fragrances have often been ignored, except for *Citrus* [18]. Due to the critical role of aroma in plant evolution and species formation [19], isolation and identification of essential enzyme genes for terpene synthesis should not be limited to model plants and common species. Rutaceae plants with unique fragrances should be investigated.

*Zanthoxylum bungeanum* Maxim. (*Z. bungeanum*), a unique Rutaceae plant is a small tree with prickles throughout the body and an intense aroma [20]. Because of its unique flavor is widely planted in China and East Asia and is an essential economic tree species in this region [21]. According to previous studies, the monoterpenes produced by the MEP pathway are the main components of the aroma [22]. However, in addition to the isolation and identification of chemical components, *Z. bungeanum* has not yet had the functional characteristics and research of critical genes for terpene synthesis. In this study, we selected *Z. bungeanum* cv. ‘Fengxiandahongpao’ (Fengxiandahongpao) as plant material to study *ZbDXS* and *ZbFPS*, which belong to the MEP pathway’s upstream rate-limiter enzymes and the MEP downstream rate-limiter enzymes of the MVA pathway. We performed molecular cloning and bioinformatics analysis of those two genes. The functions of genes were investigated by enzyme-linked immunosorbent assay (ELISA). Furthermore, to further verify their function, we constructed a transient expression vector for transient overexpression in tobacco, analyzed its contribution to terpene accumulation, and investigated their expression patterns in different tissues. We hope that these results can supplement the study of plant terpene synthesis pathways and provide references for the research of Rutaceae plants, at the same time providing insights for genetic breeding and improving the quality of *Z. bungeanum*.

## 2. Materials and Methods

### 2.1. Plant Materials

The plant material used in this experiment is *Z. bungeanum* cv. ‘Fengxiandahongpao’, was obtained from the Research Centre for Engineering and Technology of *Zanthoxylum*, State Forestry Administration, Northwest A&F University in Fengxian, Shaanxi Province, China. Roots, stems, leaves, flowers, and fruits were harvested from three Fengxiandahongpao trees (5-year-old trees). Flowers were collected on 20 May 2019, and other tissues were collected on 20 July 2019. During sampling, all samples were quickly placed in liquid nitrogen and then transported to the laboratory for storage at −80 ℃ until use. Five tissues (each tissue was sampled at 300 mg) of three random trees were collected as biological repeats.

### 2.2. RNA Extraction and cDNA Synthesis

The RNAprep Pure Plant Plus Kit (TIANGEN, Beijing, China) extracted RNA from samples following the manufacturer’s instructions. The purity and concentration of the RNA were detected by an ultra-micro nucleic acid analyzer (Nano200, AoSheng, Hangzhou, China). cDNA was obtained by reverse transcription by *EasyScript*^®^ One-Step gDNA Removal and cDNA Synthesis SuperMix (TransGen Biotech, Beijing, China) according to the manufacturer’s specifications.

### 2.3. Gene Cloning and Sequence Analysis

Full-length coding sequences (CDSs) of *ZbDXS* and *ZbFPS* were cloned using primers described in Appendix A. Using the previously reported transcriptome database as a reference [23], the obtained sequences were compared with the sequences of related species using the BLASTX search of National Center for Biotechnology Information (NCBI) data to determine sequence integrity. Concerning *ZbFPS* and *ZbDXS* gene sequences, primers with the homologous arm of pET-28a were designed (Appendix A), and high-fidelity DNA polymerase (NEB, Beijing, China) was used to amplify the gene. Gel recovery was made using the TIANgel Purification Kit (TIANGEN, Beijing, China). The target gene was ligated into the pUC19 control vector and transformed into *E. coli* TOP10 competent cells. The cells were spread on LB medium, cultured overnight, and a monoclonal for PCR identification was selected before sequencing.

### 2.4. Recombinant Protein and Enzymatic Activity Assay

Homologous recombination vector pET-28a construction was performed using a seamless cloning kit (TransGen Biotech, Beijing, China). The cutting site was selected, and the pET-28a vector was double digested with *Hind III* and *EcoR I*. The recombinant vectors *pET-28a-ZbFPS* and *pET-28a-ZbDXS* were verified to be correct after sequencing and transformed into *E. coli* BL21 Star competent cells (Biomed, Beijing, China). The resulting positive colonies were inoculated into LB (Kana) liquid medium and cultured overnight at 37 °C with shaking. The obtained bacterial solution was mixed with 30% glycerol in a ratio of 1: 1and stored at −80 ℃. The glycerol bacteria (1 μL) was diluted by 1:10,000, and 20 μL was pipetted onto kanamycin (Kana) LB solid medium with a final concentration of 50 μg/mL and cultured at 37 °C overnight. 150 mL of LB liquid culture medium (Kana) was shaken at 37 °C until the OD600 was 0.5–0.6. Isopropyl-beta-D-thiogalactoside (IPTG) was added to a final concentration of 0.5 mmol/L to induce the recombinant protein. The bacterial solution without IPTG was used as a control and centrifuged at 8500 rpm for 7 min to collect bacteria and detect protein induction. The collected bacterial cells were resuspended in 1 × phosphate-buffered saline (PBS). The bacterial solution was sonicated for 20 min in an ice bath. The supernatant was collected by centrifugation at 12,000 rpm for 15 min, and then filtered through a 0.22-micron filter. HisTrap FF (prepacked with precharged Ni Sepharose ™ 6 Fast Flow, GE, USA) was used to purify the supernatant according to the manufacturer’s instructions. The protein eluate was dialyzed using a dialysis bag at 4 ℃, and imidazole was removed. The column was washed with 5 column volumes of distilled water, then stored at 4 °C. Sodium dodecyl sulfate-polyacrylamide gel electrophoresis (SDS-PAGE) (12%) was used for analysis. A plant *DXS/FPS* kit (mobile, Shanghai, China) was used to measure the enzyme activity based on ELISA according to the manufacturer’s instructions. Three biological assays were conducted.

### 2.5. Transient Overexpression in Tobacco Leaves

Transient expression was performed in tobacco, for which *Nicotiana benthamiana* were planted in a greenhouse (28 ± 2 °C in the day, 25 ± 2 °C at night). The transient expression vector pBI121 was constructed using a pEASY^®^-Basic Seamless Cloning and Assembly Kit (TransGen Biotech, Beijing, China) following the manufacturer’s instructions. According to the manufacturer’s instruction, the constructed vector was transferred into GV3101 *Agrobacterium* electrocompetent cells (Biomed, Beijing, China) for *Agrobacterium* infection [24]. The transient expression tobacco expressing pBI121 was used as the control. 6–8 cotyledons were infected, and the injection area was marked after injection. After 72 h of cultivation, the leaves in this area were immediately frozen in liquid nitrogen. After that, the samples were stored at −80 ℃ for volatile detection by HS-SPME–GC-MS. All experiments were performed in triplicate.

### 2.6. Volatile Analysis by HS-SPME–GC-MS

VOCs in tobacco were collected by Headspace solid-phase microextraction (Agilent, Santa Clara, CA, USA). The samples were quickly ground by liquid nitrogen, and then 0.1 g of the sample was weighed into a sample bottle [22]. They were maintained at 60 °C for 30 min and adsorbed by 65m PDMS/DVB fiber. Next, the sample bottle was placed in the sample inlet and maintained at 60 ℃ for 30 min while waiting for GC-MS (Thermo Fisher Scientific ISQ&TRACE1310, Thermo Scientific, Waltham, MA, USA) analysis. The chromatographic conditions were: TG-5MS column (30 m × 0.25 mm, 0.25 m; Thermo Scientific, Waltham, MA, USA). The experimental procedure was as follows [25]. The initial temperature was 50 °C, held for 2 min, and then it was heated up to 130 °C at 3 °C min^−1^ for 2 min. The temperature was increased to 200 °C at 4 °C min^−1^ for 2 min, then increased to 230 °C at 20 °C min^−1^ for 5 min. The carrier gas was helium with a flow rate of 80 mL min^−1^. The MS conditions included ionization energy of 70 eV, and the mass scan range was 50–550 m/z. The mass spectral library of the NIST was used to identify the VOCs. All experiments were performed in three biological replicates.

### 2.7. Real-Time Quantitative PCR

The *ZbDXS* and *ZbFPS* primers were designed by Primer Premier 5.0 (Palo Alto, CA, USA) and are described in Appendix A. qRT-PCR was performed on a CFX96 Real-Time System (Bio-Rad Laboratories, Inc., Hercules, CA, USA) with using SYBR^®^ Green Premix Pro Taq HS qPCR Kit (Accurate Biotechnology Co., Ltd., Hunan, China). *ZbUBQ* and *ZbUBA* were used as housekeeping reference genes [26]. The PCR procedure was as follows: 95 °C for one cycle of 3 min, followed by 40 cycles, including 95 °C for 15 s; 60 °C for 15 s; and 72 °C for 30 s. Finally, the reaction was terminated with 95 °C for 10 s; 65 °C for 5 s. The method 2^−^^△△^^Ct^ was used to calculate the relative mRNA abundance. All experiments were performed in three biological replicates.

### 2.8. Bioinformatics Analysis

MEGA 6.0 software (Center for Evolutionary Medicine and Informatics, Tempe, AZ, USA) was used to perform phylogenetic analysis. Prot Param (https://web.expasy.org/protparam/) (accessed on 12 May 2019)was used to analyze the basic properties of protein sequences. Prot Scale (http://web.expasy.org/protscale/) (accessed on 19 May 2020) was used to predict the hydrophobicity/hydrophilicity of amino acid sequences. SignalP 4.1 Server (http://www.cbs.dtu.dk/services/SignalP-4.1/) (accessed on 22 May 2020) was used to predict signal peptides of amino acid sequences. Plant-Ploc (http://www.csbio.sjtu.edu.cn/cgi-bin/PlantPLoc.cgi) (accessed on 23 November 2019) was used for subcellular localization prediction. SWISS-MODEL (http://swissmodel.expasy.org/) (accessed on 27 May 2020) was used to analyze the 3D homologous modeling. The multi-sequence alignment of amino acid sequences was performed using DAMAN (LynnonCorporation, Quebec, Canada) software.

## 3. Results

### 3.1. Molecular Cloning and Sequences Analysis

This experiment cloned CDSs of *ZbDXS* (MT364247) and *ZbFPS* (MT364248). The *ZbDXS* CDS was 2172 base pairs (bp) in length and encoded a peptide of 723 amino acids, while the *ZbFPS* CDS was1029 bp and encoded a peptide of 342 amino acids. Two fragments were amplified by RT-PCR, one about 1000 bp and the other about 2100 bp. These results were consistent with expectations (Appendix A). BLAST performed an analysis of the deduced amino acid sequence of *ZbDXS* and *ZbFPS*. It showed a high degree of sequence similarity of *Zb**DXS* with other plant species, such as *Citrus sinensis* (XP_006489186.1, 94.19%, identity), *Pistacia vera* (XP_031247814.1, 89.10%, identity) and *Herrania umbratica* (XP_021291622.1, 88.95%, identity) (Appendix A). As for *ZbFPS*, it had 94.74, 85.67, and 85.09% identity with FPSs from *Citrus clementina* (XP_006452657.1), *Coffea arabica* (XP_027126964.1), and *Gardenia jasminoides* (AYC62333.1) (Appendix A).

### 3.2. Bioinformatics Analysis

To further clarify the evolutionary status of *ZbFPS* amino acid sequences, a phylogenetic tree was established by the neighbor-joining method (we have clarified the bioinformatics analysis of *ZbDXS* in previous studies) [22]. At the tip of the clade, the shorter branch length proves the vital amino acid conservation of its clustering genes and illustrates the close evolutionary relationship between members. The longer branch length indicates that its members evolve faster [27]. According to Figure 1, all plant FPS sequences were separated into three main groups. *ZbFPS* was most closely related to *CcFPS* (*Citrus clementina*) and *CsFPS* (*Citrus sinensis*), both of which belong to the family Rutaceae. It was speculated that *Z. bungeanum* and *citrus* may be derived from the same ancestor. *Z. bungeanum* had a faster evolutionary speed than *citrus*, and *citrus* genes had more vital amino acid conservation than *Z. bungeanum.* At the same time, according to the relationship, *ZbFPS* had a close kinship with *MiFPS* (*Mangifera indica*), *DzFPS* (*Durio zibethinus*), *RrFPS* (*Rosa rugosa*), and *SgFPS* (*Siraitia grosvenorii*), which were primarily trees with prickles and aroma. The amino acid sequence of the phylogenetic tree was determined from NCBI.

Prot Param analyzed the protein information of *ZbFPS*, and the results were shown in Appendix A. Prot Scale was used to predict the hydrophobicity/hydrophilicity of the amino acid sequences of *ZbFPS* (Appendix A). For *ZbFPS*, position 309 of the polypeptide chain was the most hydrophilic with the lowest score of −2.867, and position 202 was the most hydrophobic with the highest score of 2.444. Overall, the distribution of hydrophilic amino acids was relatively uniform, and the number was more significant than that of hydrophobic amino acids. Therefore, it was speculated that the *ZbFPS* protein was a hydrophilic protein. Prediction of the signal peptide of the amino acid sequence of *ZbFPS* was made using Signal P 4.1 Server (Appendix A). Generally, the mean S value (mean S-score) was used to consider whether it was a secreted protein. The average value of S was more significant than 0.5, which was predicted to be a secreted protein and the presence of a signal peptide. Analysis results showed that *ZbFPS* was not secreted by proteins. Plant-mPLoc and Euk-mPLoc 2.0 were used for subcellular localization of *ZbDXS* and *ZbFPS* (Appendix A). The results showed that *ZbDXS* was most likely localized in chloroplast, and *ZbFPS* was most likely localized in mitochondria.

Under the influence of evolutionary forces, proteins form a 3D structure to meet specific functional needs. Therefore, understanding the tertiary structure of a protein can help us understand its function. However, it is time-consuming and laborious to determine the tertiary structure of the protein through experiments, and building a model through a computer can make it easier for us to understand the properties of the protein [28]. The predicted tertiary structures of *ZbDXS* and *ZbFPS* are shown in Figure 2. Human *DXS* with 41.34% sequence identity served as a template for comparative modeling. The GMQE (Global Model Quality Estimation) was >0.63 (Appendix A), and the QMEAN (Qualitative Model Energy ANalysis) was>−3.04 (Appendix A). The 3D model for *ZbFPS* was established based on the template of human *FPS* (82.06% sequence identity). The GMQE was >0.96, indicating the high quality of the result, and the QMEAN was >−0.06 (Appendix A). The predicted structures of *ZbDXS* and *ZbFPS* consisting predominantly of alpha-helices were folded as a single domain.

### 3.3. Protein Purification, Induction, and In Vitro Enzymatic Activity Testing

*ZbDXS* and *ZbFPS* were cloned and transferred to *E. coli* for prokaryotic expression analysis to study the relationship between genes and terpene metabolism. IPTG-induced bacteria were analyzed by 12% SDS-PAGE. The results showed that the two specific target fragments are 76 kDa and 40 kDa, respectively, consistent with the expected sizes of *ZbDXS* and *ZbFPS* (Figure 3). Based on the precipitation and supernatant results, it was speculated that the recombinant proteins *ZbDXS* and *ZbFPS* existed as inclusion bodies. To determine whether *ZbDXS* and *ZbDXR* have a function, the purified proteins were preliminarily determined by enzyme activity. *ZbDXS* and *ZbFPS* enzymes were determined with the enzyme-linked reaction kit. The purified *ZbDXS* and *ZbFPS* captured antibodies were coated with microporous plates to make solid-phase antibodies. The purified proteins were successively added to the coated micropores and then bound with horseradish peroxidase (HRP)-labeled detection antibodies to form an antibody-antigen-enzyme-labeled antibody complex. After thorough washing, the substrate 3, 3′, 5, 5′-Tetramethylbenzidine (TMB) was added for color development. TMB was converted to blue under the catalysis of the HRP enzyme and the final yellow under the action of acid. The depth of color was positively correlated with the purified protein in the sample. Absorbance was determined at 450 nm wavelength by an enzyme marker. The standard curve calculated the enzyme activity in the sample. As shown in Table 1, both of the two purified proteins had enzyme activity, and the enzyme activity of *ZbFPS* was higher than that of *ZbDXS*. Both genes had functions of *DXS* and *FPS*.

### 3.4. Transient Overexpression of ZbDXS and ZbFPS Increased Production of Terpene

To understand further the role of *ZbDXS* and *ZbFPS* in terpene synthesis, two genes were used for functional analysis via *Agrobacterium*-mediated transformation in *Nicotiana benthamiana*. Infiltrated tissues were sampled 72 h post infiltration and subjected to GC-MS analysis. GC-MS results showed that the overexpression of *ZbDXS* and *ZbFPS* increased the monoterpene content in *Nicotiana benthamiana* (the content of D-limonene and linalool increased by about two times), which indicated that *ZbDXS* and *ZbFPS* can promote the synthesis of terpene (Figure 4).

### 3.5. Expression Profile of ZbDXS and ZbFPS in Different Tissues

To investigate the expression patterns of *ZbDXS* and *ZbFPS*, the *Z. bungeanum* tissue-specific expression pattern was assessed (Figure 5). RT-qPCR analysis showed that two genes had different expression patterns in different tissues of *Z. bungeanum*. In general, both genes were expressed in all tissues. *ZbDXS* had the highest relative expression in fruits, followed by leaves, stems, and flowers; *ZbFPS* had the highest relative expression in fruits, followed by leaves and flowers, and the lowest relative expression in stems (Appendix A). It was noteworthy that both genes were highly expressed in the fruit, which may be related to the relatively high level of terpene accumulation in the fruit, which required further study.

## 4. Discussion

Terpenes are a component of plant essential oils [29]. Terpenes are closely related to human life and are widely used in pesticides, perfumes, nutraceuticals, and medicines [30]. Recently, it was reported that terpenes have great potential as biofuels [31]. Due to the great potential of terpenes, Rutaceae plants with developed oil glands and abundant terpenes are worthy of study. Here, we focus on two key genes (*ZbDXS, and ZbFPS*) in the terpenoid metabolic pathway of *Z. bungeanum*. The current research testifies that *ZbDXS* and *ZbFPS* are related to terpene biosynthesis, which is based on evidence from an in vitro assay of a recombinant protein and transient expression in tobacco.

According to the phylogenetic tree, we found that the *FPS* gene is mainly distributed in gymnosperms, monocotyledons, and dicotyledons. The evolution of genes leads to functional differentiation, which in turn affects the diversity of metabolism. Genetic relationships between different species can be determined by phylogenetic tree analysis of the terrestrial plants of the *FPS* gene [32]. According to the phylogenetic tree lineage, it was speculated that the metabolites of different species differ due to their metabolic pathways. The terrestrial plant’s *FPS* gene is located at a branch point in the terpene pathway, directing carbon out of the central part of the isoprene pathway [33]. Subcellular localization prediction showed that *ZbDXS* was located in the chloroplast, which was consistent with the MEP pathway occurring in the chloroplast. Notably, *ZbFPS* were localized in mitochondria, consistent with previous reports [34,35,36,37,38]. Current studies have found that *FPS* genes are located in chloroplasts, cytosol, and peroxisomes [39,40,41]. We deduce that *ZbFPS* may be transported from the cytoplasm to mitochondria, through which energy is provided for downstream product synthesis.

When plants are under stress, the terpenes pathway responds to environmental factors to stress resistance, thereby protecting plants [16]. The interaction of upstream and downstream genes in the terpene pathway forms a feedback regulation mechanism, the core of coping with stress. The interactions of *ZbDXS* and *ZbFPS* with other regulatory proteins were predicted by STRING (http://string-db.org/) (accessed on 29 May 2020). Input *ZbDXS* and *ZbFPS* into the STRING website to predict protein interactions. The specie selection was the *Arabidopsis thaliana* for analysis. Among them, *ZbDXS* matches *DXPS1* (73%), and *ZbFPS* matches *FPS2* (80%). We can find from the protein functional connection network that *ZbDXS* and *ZbFPS* have complex interactions with various proteins (Figure 6). These proteins are involved in the terpene pathway, where *ISPF* and *DXR* are at the center of the functional connection network and interact with various proteins. The results showed that genes in the terpene pathway interact with each other, antagonizing each other and producing terpenes in response to stress response.

Tissue expression could be used to regulate the supply to biosynthetic pathways localized in these plant parts [14]. The *DXS* family is involved in the biosynthesis of different isoprene compounds due to its different members [42]. *DXS*
*1* is expressed in many plant tissues except the roots, such as corn, tomato, and so on. *DXS*
*2* is strongly expressed in mycorrhizal-induced roots and accumulates carotenoids and monoterpenes [12]. *FPS* is also expressed differently in different tissues of the plant. For *Matricaria recutita* L., *MrFPS* is most highly expressed in flowers and stems [43]. In *H. brasiliensis*, *HbFPS* is not expressed in roots, stems, leaves, and other tissues but is highly expressed in latex [44]. These studies have shown that the spatiotemporal expression patterns of *ZbDXS* and *ZbFPS* genes in different plants are quite different. *ZbDXS* and *ZbFPS* have different expressions in various tissues of *Z. bungeanum.* Compared with other tissues, both *ZbDXS* and *ZbFPS* have the highest expression in fruits. While *ZbDXS* has the lowest expression in the stem, and *ZbFPS* has the lowest expression in the root. The high expression of *ZbDXS* and *ZbFPS* in fruits is consistent with their transcription and accumulation of volatile oils, indicating that they play a role in terpene precursor biosynthesis. The role of *DXS* and *FPS* in the biosynthesis of other isoprenoids, especially the analysis of the upstream and downstream pathways, needs further research.

In addition, due to the genetic transformation system of *Z.*
*bungeanum* being more difficult than herb plants, such as *Arabidopsis* and *Oryza sativa,* the growth cycle of *Z.*
*bungeanum* is longer and transient expression is an effective means to study the function. This study performed transient tobacco expression experiments with *ZbDXS* and *ZbFPS.* Compared with the control group, overexpression of both *ZbDXS* and *ZbFPS* increased the terpene content in tobacco. Contents of D-limonene and linalool increased by about two times. We provide a piece of evidence for the active participation of *ZbDXS* and *ZbFPS* in the synthesis of terpenes. Overexpression of *ZbDXS* and *ZbFPS* may affect genes upstream or downstream of metabolic pathways, thereby altering the flux of isoprene. Although overexpression of *ZbDXS* and *ZbFPS* increased terpene content, they did not change the composition of terpenes in tobacco.

In conclusion, we cloned two CDSs of *ZbDXS* and *ZbFPS*, which are involved in terpene biosynthesis in *Z. bungeanum*. The catalytic functions of the enzymes encoded by these genes were identified. Analysis of the expression patterns of the two genes shows that these two genes have different expression patterns. We provide new effective strategies and research methods for improving the quality and utilization of germplasm resources of *Z. bungeanum* and provide new ideas for the research of Rutaceae.

## 5. Conclusions

In this experiment, we cloned the CDSs of *DXS* and *FPS* genes in *Z. bungeanum*. The CDS sequence length of the cloned *ZbDXS* was 2172 bp, encoding 723 amino acids. The CDS sequence length of *ZbFPS* was 1029 bp, encoding 342 amino acids. Bioinformatics analysis showed that *ZbDXS* was most likely to be localized in chloroplast, and *ZbFPS* was most likely localized in mitochondria. *ZbFPS* is a hydrophilic protein but not secreted protein. Phylogenetic tree analysis showed that *ZbFPS* has the closest relationship with *Mangifera indica*, *Durio zibethinus*, *Rosa rugosa,* and *Siraitia grosvenorii*. In vitro enzyme activity experiments showed that the purified proteins *ZbDXS* and *ZbFPS* had the functions of the *DXS* enzyme and the *FPS* enzyme. The transient expressions of *ZbDXS* and *ZbFPS* in the tobacco leaves increased the terpene content of tobacco. The relative expression of *ZbDXS* and *ZbFPS* was the highest in the fruit, among which the relative expression of *ZbDXS* was the lowest in the root, and the relative expression of *ZbFPS* was the lowest in the stem.

## Figures and Tables

**Figure 1 foods-11-01746-f001:**
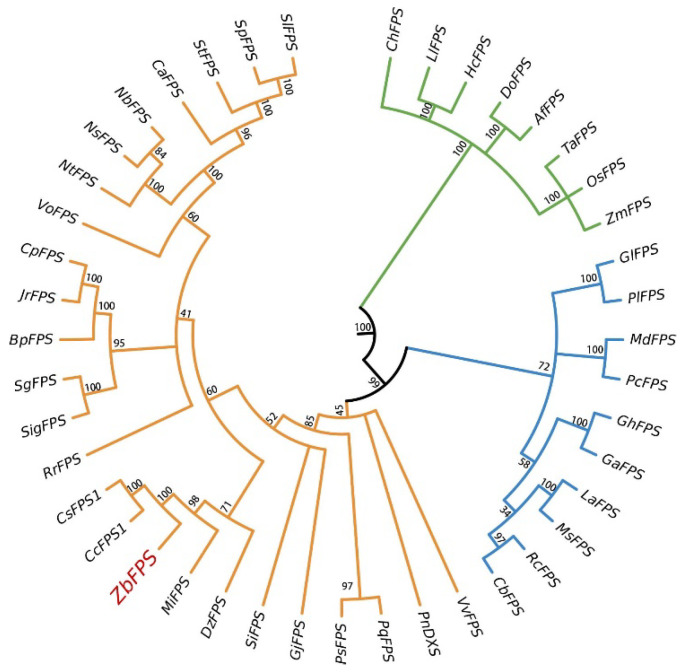
Results of phylogenetic analysis of ZbFPS and other plants. The numbers in the figure represent the genetic distance. Bootstrap values are shown as a percentage of 1000 replicates. Different colors represent different clades. ZbFPS is marked by green dots. The GeneBank sequence number of the related sequence: *Citrus sinensis CsFPS1* (XM_025097884.1); *Citrus clementina CcFPS1* (XM_006452594.2); *Mangifera indica MiFPS* (JN035296.1); *Betula platyphylla BpFPS* (KP729177.1); *Panax notoginseng PnFPS* (DQ059550.1); *Durio zibethinus DzFPS* (XM_022898165.1); *Vitis vinifera VvFPS* (NM_001280935.1); *Cyclocarya paliurus CpFPS* (GU121224.1); *Siraitia grosvenorii SgFPS* (KJ139983.1); *Valeriana officinalis VoFPS* (KP735611.1); *Chlorophytum borivilianum CbFPS* (KM272205.1); *Chimonanthus praecox ChFPS* (FJ415102.1); *Lilium longiflorum LlFPS* (JF273657.1); *Juglans regia JrFPS* (XM_018957967.1); *Medicago sativa MsFPS* (GU361537.1); *Ricinus communis RcFPS* (KT306005.1); *Pyrus communis PcFPS* (KF855953.1); *Malus domestica MdFPS* (NM_001293873.1); *Paeonia lactiflora PlFPS* (KP708571.1); *Nicotiana tabacum NtFPS* (XM_016630722.1); *Nicotiana benthamiana NbFPS* (LC015753.1); *Nicotiana sylvestris NsFPS* (XM_009766390.1); *Solanum tuberosum StFPS* (XM_006344841.2); *Solanum lycopersicum SlFPS* (NM_001247139.2); *Solanum pennellii SpFPS* (XM_015206329.1); *Capsicum annuum CaFPS* (NM_001324870.1); *Panax sokpayensis PsFPS* (KT936527.1); *Panax quinquefolius PqFPS* (KC524468.1); *Sesamum indicum SiFPS* (XM_011099429.1); *Zea mays ZmFPS* (NM_001111569.1); *Lupinus albus LaFPS* (U15777.1); *Gossypium arboreum GaFPS* (NM_001330011.1); *Oryza sativa OsFPS* (D85317.1); *Anoectochilus formosanus AfFPS* (MH104946.1); *Gardenia jasminoides GjFPS* (MG811940.1); *Gossypium hirsutum GhFPS* (NM_001326961.1); *Rosa rugosa RrFPS* (KP768082.1); *Triticum aestivum TaFPS* (JX235717.1); *Hedychium coccineum HcFPS* (JN695015.1); *Dendrobium officinale DoFPS* (JX679465.1); *Gentiana lutea GlFPS* (EF203252.1); *Siraitia grosvenorii SigFPS* (KJ139983.1).

**Figure 2 foods-11-01746-f002:**
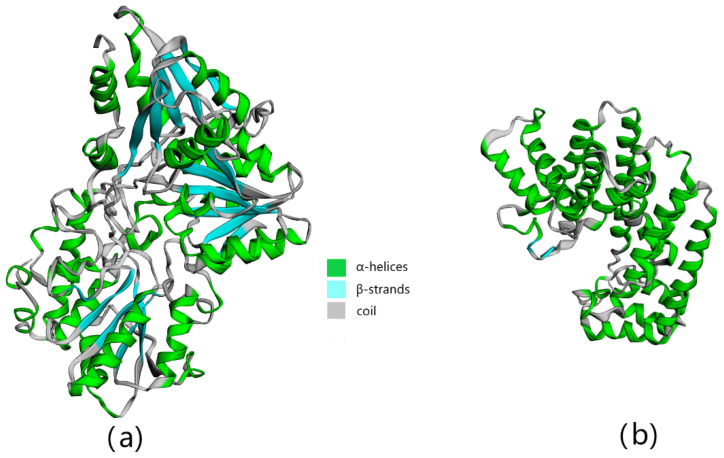
3D constitution prediction of the *ZbDXS* (**a**) and *ZbFPS* (**b**) amino acid sequence.

**Figure 3 foods-11-01746-f003:**
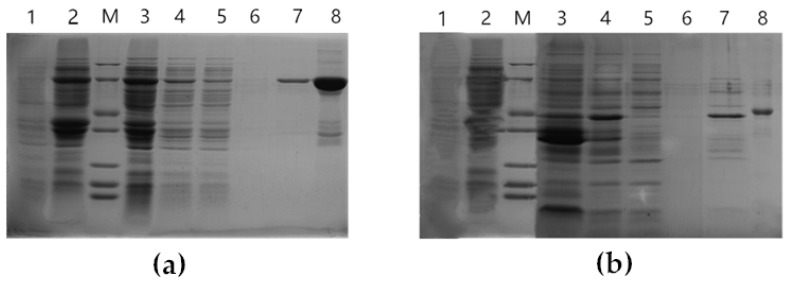
Analysis of expressed fusion proteins containing *ZbDXS* (**a**) and *ZbFPS* (**b**). Lane 1: uninduced; Lane 2: total proteins from IPTG-induced; Lane M: protein marker; Lane 3: proteins from the sediment; Lane 4: proteins from the supernatant; Lane 5: proteins from the flow-through; Lane 6: proteins from the first elution; Lane 7: proteins from the second elution; Lane 8: Concentrated protein.

**Figure 4 foods-11-01746-f004:**
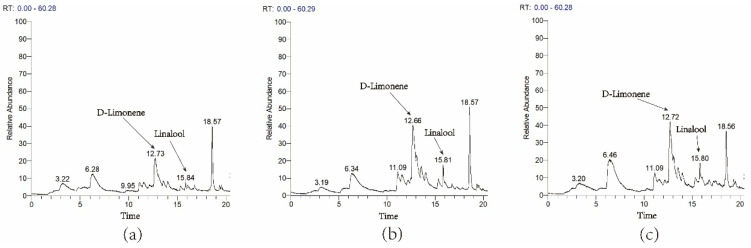
Transient overexpression of *ZbDXS* and *ZbFPS* in *Nicotiana benthamiana*. Control group (**a**). Transient expression of *ZbDXS* in *Nicotiana benthamiana* (**b**). Transient expression of *ZbFPS* in *Nicotiana benthamiana* (**c**). The *X*−axis represents the retention time of the peak outflow, and the *Y*-axis represents the relative integrated area of the chromatographic peak.

**Figure 5 foods-11-01746-f005:**
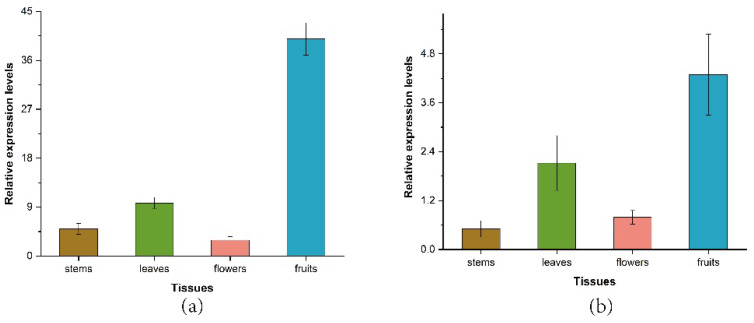
Expression analysis of the *ZbDXS* and *ZbFPS* genes in various tissues. *ZbDXS* expression levels in tissues (**a**). *ZbFPS* expression levels in tissues (**b**). Data presented are 2^−ΔΔCt^ levels calculated relative to the special tissue (root). Data are presented as mean ± SE, *n* = 3.

**Figure 6 foods-11-01746-f006:**
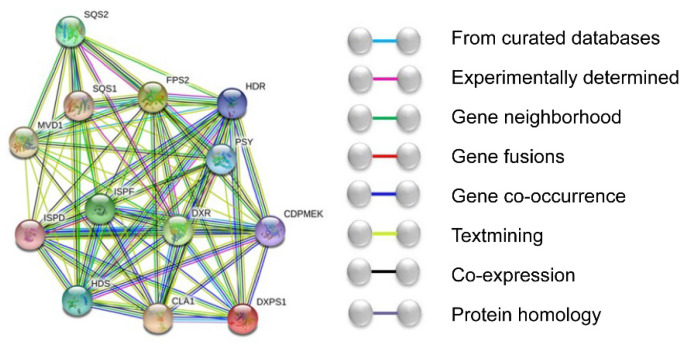
The *ZbDXS* and *ZbFPS* protein functional connection network.

**Table 1 foods-11-01746-t001:** Analysis of *ZbDXS* and *ZbFPS* in vitro ELISA.

Gene Name	Substrates	Content(U/L) ± SD
*ZbDXS*	TMB	0.29 ± 0.01
*ZbFPS*	TMB	0.73 ± 0.03

Note: The data were of three biological replicates, and SD represents the standard deviation.

## Data Availability

Data is contained within the article or Appendix A.

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
