# Peer review of "Molecular Cloning and Functional Analysis of DXS and FPS Genes from Zanthoxylum bungeanum Maxim"

_foods, 2022, doi:10.3390/foods11121746_

Round 1

Reviewer 1 Report

The manuscript:” Molecular Cloning and Functional Analysis of DXS and FPS Genes from Zanthoxylum bungeanum Maxim.” is dealing with cloning of terpene metabolic gene. Coding sequence of two enzymes were cloned from Z. bungeanum cv. ‘Fengxiandahongpao’(1-deoxy-D-xylulose-5-phosphate synthase, DXS and farnesyl pyrophosphate synthase, FPS). ZbDXS and ZbFPS were expressed in different tissues of Z. bungeanum with highest relative expression in fruits. Transient expression of ZbDXS and ZbFPS in tobacco significantly increased the terpene content. Authors suggested that ZbDXS and ZbFPS are positively related to terpene synthesis. The manuscript has some merit, falls in a scope of the journal Foods.

Here are some suggestions for manuscript improvement. All misunderstandings and errors are highlighted in manuscript text.

Line 45: Correct the word

Line 91: Why jasmonic acid underline?

Line 109: thoes?

Line 224: Why experiment underline?

Line 317: Correct the words

Line 323: table with capital letter?

Line 333: SD is usually abbreviation of Standard Deviation

Line 336: Agrobacterium in italic?

Line 367: Correct please.

Line 381/382: Is this sentence correct?

References 47, 48: There is confusion with references 47 and 48. In reference 47 there is no data about Matricaria resituta and there is no reference 48 in reference list.

Author Response

Dear Editor and Reviewers,

Thanks very much for taking your time to review this manuscript. I really appreciate all your comments and suggestions!

I have revised all comments one by one for your review and highlight in the article. These include your proposed additions to the references and so on.

Kind regards

 Tian Lu

Reviewer 2 Report

 My comments are below.

In reviewed manuscript, authors studied coding sequence of 1-deoxy-D-xylulose-5-phosphate synthase (DXS) and farnesyl pyrophosphate synthase (FPS) were cloned from Z. bungeanum cv. ‘Fengxiandahongpao’. Further bioinformatics results showed deduced ZbFPS were hydrophilic proteins and do not contain transmembrane domains. Furthermore, authors showed purified proteins ZbDXS and ZbFPS functions of DXS enzyme and FPS enzyme. The transient expression of  ZbDXS and ZbFPS in tobacco both significantly increased the terpene content of tobacco. Moreover, ZbDXS and ZbFPS were expressed in different tissues of Z. bungeanum, and the relative expression  of the two genes was the highest in fruits. In general, the manuscript represents very big piece of research in logical presentation. Therefore, it might be conditionally accepted with a subject to major revision. However, this major revision means only that there is no necessity to repeat or extend the experiments and their analysis. Instead, authors have to improve their manuscript with many non-clear meaning, inaccuracy and inconsistency, and the authors need to address the following issues.

  1. I have read the entire manuscript and my initial comment is that manuscript is poorly written and needs to reflect the significant finding or novelty.

  1. I have significant concerns about the grammar and vocabulary of the manuscript; therefore, the improvement of the language is highly needed.

  1. The structure of the abstract should be improved, as well as the lack of several aspects that should be included in this section. Most of the abstract contains confusing and uninformative sentences. Please give more precise objectives here (such as in the Abstract) and put the main finding of study in abstract.

  1. Introduction grammatical issues appear to be most prevalent in the introduction, making for very confusing reading. Further, the introduction is long but has no clear thread.

  1. General note: the figures in this section are quite low resolution and difficult to make out. Higher-resolution versions will be needed for publication, for example Figure 1, Figure 4, Figure 6, Figure S2, Figure S4.

  1. In supplementary S2 replace with Figure S2.

  1. Supplements in the case of Figures it is necessary to add a legend so that the figures are self-explanatory.

  1. qRT-PCR methodology provided is also very vague and confusing. Please provide more details like what was the calibrator used in the study. I assume the authors have used the control as the calibrator. If so, the authors should not include the control within the bar graph as it represents the fold change between the treated vs control and a fold change of “1” for the ‘control’ doesn’t make any sense.  Also, would be good to provide details on what reagents (details of probes used, if any, if SYBR was used then details for that, etc.) and real time PCR machine were used in the current study.
  2. Discussion - many times references are made to the information given in the Introduction section (sometimes more general information). It would be good to discuss especially the results and critically, ie. Which can cause differences in the results of authors and other articles.

  1. The authors have not followed the journal format in the reference section, for instance, the Journal name should be abbreviated.

Author Response

Dear Editor and Reviewers,

Thanks very much for taking your time to review this manuscript. I really appreciate all your comments and suggestions! All changes have been highlighted in the PDF version of the manuscript.

1.I have read the entire manuscript and my initial comment is that manuscript is poorly written and needs to reflect the significant finding or novelty.

Thank you for the suggested,in order to highlight significant finding, the abstract and introduction have been revised in many ways for your review.

2.I have significant concerns about the grammar and vocabulary of the manuscript; therefore, the improvement of the language is highly needed.

Thank you for the suggested,in order to improve the language, the full text has been professionally polished by MDPI.

3.The structure of the abstract should be improved,as well as the lack of several aspects that should be included in this section. Most of the abstract contains confusing. and uninformative sentences. Please give more precise objectives here (such as in the Abstract) and put the main finding of study in abstract. written and needs to reflect the significant finding or novelty.

Thank you for the suggested,the summary has been carefully revised to add logic and critical point findings for your review.

  1. Introduction grammatical issues appear to be most prevalent in the introduction, making for very confusing reading. Further, the introduction is long but has no clear thread.

Thank you for the suggested,the unimportant parts in the introduction have been deleted, and the grammar has been modified, please review.

5.General note: the figures in this section are quite low resolution and difficult to make out. Higher-resolution versions will be needed for publication, for example Figure 1, Figure 4, Figure 6, Figure S2, Figure S4.

Thank you for the suggested,clearer illustrations have been reworked.

6.In supplementary S2 replace with Figure S2.

Thank you for the suggested,supplementary has been rearranged and edited.

7.Supplements in the case of Figures it is necessary to add a legend so that the figures are self-explanatory.

Thank you for the suggested,in the Supplements, part of the image is displayed on the second page due to the size of the image, which has been adjusted to be displayed on the first page.

8.qRT-PCR methodology provided is also very vague and confusing. Please provide more details like what was the calibrator used in the study. I assume the authors have used the control as the calibrator. If so, the authors should not include the control within the bar graph as it represents the fold change between the treated vs control and a fold change of“1” for the‘control’ doesn’t make any sense. Also, would be good to provide details on what reagents (details of probes used, if any, if SYBR was used then details for that, etc.) and real time PCR machine were used in the current study.

Thank you for the suggested,We have added more detailed steps and methods for QRT-PCR. We fully agree with your opinion on“1”  and have made modifications in Figure 5.

9.Discussion - many times references are made to the information given in the Introduction section (sometimes more general information). It would be good to discuss especially the results and critically, ie. Which can cause differences in the results of authors and other articles.

Thank you for your valuable advice. We agree with you and have made optimization.

10.The authors have not followed the journal format in the reference section, for instance, the Journal name should be abbreviated.

Thank you for your correction. This obvious mistake is our oversight. We are extremely sorry Thanks for your time.

Kind regards

 Tian Lu

Round 2

Reviewer 2 Report

Dear Editor,

Thank you for providing the opportunity to review the revised manuscript. . The manuscript is improved considerably after revision according to the reviewer's comment. Now this study is a suitable contribution to the Food.

Thank you

With best regards

Author Response

Dear  Reviewers

Thanks very much for taking your time to review this manuscript. I really appreciate all your comments and suggestions! 

Thank you for your correction. Thanks for your time.

Kind regards

 Tian Lu